# KANERVA++: EXTENDING THE KANERVA MACHINE WITH DIFFERENTIABLE, LOCALLY BLOCK ALLOCATED LATENT MEMORY

**Jason Ramapuram**[1,2]**, Yan Wu**[3]**, Alexandros Kalousis**[1]

[1] University of Geneva & Geneva School of Business Administration, HES-SO
[2] Currently at Apple
[3] Deepmind
jramapuram@apple.com, yanwu@google.com, alexandros.kalousis@hesge.ch

## ABSTRACT

Episodic and semantic memory are critical components of the human memory model. The theory of complementary learning systems (McClelland et al., 1995) suggests that the compressed representation produced by a serial event (episodic memory) is later restructured to build a more generalized form of reusable knowledge (semantic memory). In this work we develop a new principled Bayesian memory allocation scheme that bridges the gap between episodic and semantic memory via a hierarchical latent variable model. We take inspiration from traditional heap allocation and extend the idea of locally contiguous memory to the Kanerva Machine, enabling a novel differentiable block allocated latent memory. In contrast to the Kanerva Machine, we simplify the process of memory writing by treating it as a fully feed forward deterministic process, relying on the stochasticity of the read key distribution to disperse information within the memory. We demonstrate that this allocation scheme improves performance in *memory conditional* image generation, resulting in new state-of-the-art conditional likelihood values on binarized MNIST ($\leq$**41.58 nats/image**) , binarized Omniglot ($\leq$**66.24 nats/image**), as well as presenting competitive performance on CIFAR10, DMLab Mazes, Celeb-A and ImageNet32$\times$32.

## 1 INTRODUCTION

Memory is a central tenet in the model of human intelligence and is crucial to long-term reasoning and planning. Of particular interest is the theory of complementary learning systems McClelland et al. (1995) which proposes that the brain employs two complementary systems to support the acquisition of complex behaviours: a hippocampal fast-learning system that records events as episodic memory, and a neocortical slow learning system that learns statistics across events as semantic memory. While the functional dichotomy of the complementary systems are well-established McClelland et al. (1995); Kumaran et al. (2016), it remains unclear whether they are bounded by different computational principles. In this work we introduce a model that bridges this gap by showing that the same statistical learning principle can be applied to the fast learning system through the construction of a hierarchical Bayesian memory.

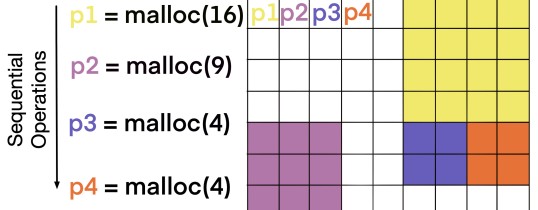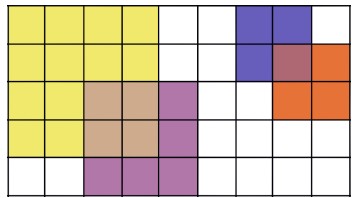

Figure 1: Example final state of a traditional heap allocator (Marlow et al., 2008) (*Left*) vs. K++ (*Right*); final state created by sequential operations listed on the left. K++ uses a key distribution to stochastically point to a memory sub-region while Marlow et al. (2008) uses a direct pointer. Traditional heap allocated memory affords $O(1)$ free / malloc computational complexity and serves as inspiration for K++ which uses differentiable neural proxies.

While recent work has shown that using memory augmented neural networks can drastically improve the performance of generative models (Wu et al., 2018a;b), language models (Weston et al., 2015), meta-learning (Santoro et al., 2016), long-term planning (Graves et al., 2014; 2016) and sample efficiency in reinforcement learning (Zhu et al., 2019), no model has been proposed to exploit the inherent multi-dimensionality of biological memory Reimann et al. (2017). Inspired by the traditional (computer-science) memory model of heap allocation (Figure 1-*Left*), we propose a novel differentiable memory allocation scheme called Kanerva ++ (K++), that learns to compress an episode of samples, referred to by the set of pointers $\{p1, ..., p4\}$ in Figure 1, into a latent multi-dimensional memory (Figure 1-*Right*). The K++ model infers a key distribution as a proxy to the pointers (Marlow et al., 2008) and is able to embed similar samples to an overlapping latent representation space, thus enabling it to be more efficient on compressing input distributions. In this work, we focus on applying this novel memory allocation scheme to latent variable generative models, where we improve the memory model in the Kanerva Machine (Wu et al., 2018a;b).

## 2 RELATED WORK

**Variational Autoencoders**: Variational autoencoders (VAEs) (Kingma & Welling, 2014) are a fundamental part of the modern machine learning toolbox and have wide ranging applications from generative modeling (Kingma & Welling, 2014; Kingma et al., 2016; Burda et al., 2016), learning graphs (Kipf & Welling, 2016), medical applications (Sedai et al., 2017; Zhao et al., 2019) and video analysis (Fan et al., 2020). As a latent variable model, VAEs infer an approximate posterior over a latent representation $Z$ and can be used in downstream tasks such as control in reinforcement learning (Nair et al., 2018; Pritzel et al., 2017). VAEs maximize an evidence lower bound (ELBO), $\mathcal{L}(X, Z)$, of the log-marginal likelihood, $\ln p(X) > \mathcal{L}(X, Z) = \ln p_\theta(X|Z) - \mathcal{D}_{KL}(q_\phi(Z|X)||p_\theta(Z))$. The produced variational approximation, $q_\phi(Z|X)$, is typically called the encoder, while $p_\theta(X|Z)$ comes from the decoder. Methods that aim to improve these latent variable generative models typically fall into two different paradigms: learning more informative priors or leveraging novel decoders. While improved decoder models such as PixelVAE (Gulrajani et al., 2017) and PixelVAE++ (Sadeghi et al., 2019) drastically improve the performance of $p_\theta(X|Z)$, they suffer from a phenomenon called posterior collapse (Lucas et al., 2019), where the decoder can become almost independent of the posterior sample, but still retains the ability to reconstruct the original sample by relying on its auto-regressive property (Goyal et al., 2017a).

In contrast, VampPrior (Tomczak & Welling, 2018), Associative Compression Networks (ACN) (Graves et al., 2018), VAE-nCRP (Goyal et al., 2017b) and VLAE (Chen et al., 2017) tighten the variational bound by learning more informed priors. VLAE for example, uses a powerful auto-regressive prior; VAE-nCRP learns a non-parametric Chinese restaurant process prior and VampPrior learns a Gaussian mixture prior representing prototypical virtual samples. On the other hand, ACN takes a two-stage approach: by clustering real samples in the space of the posterior; and by using these related samples as inputs to a learned prior, ACN provides an information theoretic alternative to improved code transmission. Our work falls into this latter paradigm: we parameterize a learned prior by reading from a common memory, built through a transformation of an episode of input samples.

**Memory Models**: Inspired by the associative nature of biological memory, the Hopfield network (Hopfield, 1982) introduced the notion of content-addressable memory, defined by a set of binary neurons coupled with a Hamiltonian and a dynamical update rule. Iterating the update rule minimizes the Hamiltonian, resulting in patterns being stored at different configurations (Hopfield, 1982; Krotov & Hopfield, 2016). Writing in a Hopfield network, thus corresponds to finding weight configurations such that stored patterns become attractors via Hebbian rules (Hebb, 1949). This concept of memory was extended to a distributed, continuous setting in Kanerva (1988) and to a complex valued, holographic convolutional binding mechanism by Plate (1995). The central difference between associative memory models Hopfield (1982); Kanerva (1988) and holographic memory Plate (1995) is that the latter decouples the size of the memory from the input word size.

Most recent work with memory augmented neural networks treat memory in a slot-based manner (closer to the associative memory paradigm), where each column of a memory matrix, $M$, represents a single slot. Reading memory traces, $z$, entails using a vector of addressing weights, $r$, to attend to the appropriate column of $M$, $z = r^T M$. This paradigm of memory includes models such as the Neural Turing Machine (NTM) (Graves et al., 2014), Differentiable Neural Computer (DNC)

(Graves et al., 2016) [1], Memory Networks (Weston et al., 2015), Generative Temporal Models with Memory (GTMM) Fraccaro et al. (2018), Variational Memory Encoder-Decoder (VMED) Le et al. (2018), and Variational Memory Addressing (VMA) (Bornschein et al., 2017). VMA differs from GTMM, VMED, DNC, NTM and Memory Networks by taking a stochastic approach to discrete key-addressing, instead of the deterministic approach of the latter models.

Recently, the Kanerva Machine (KM) (Wu et al., 2018a) and its extension, the Dynamic Kanerva Machine (DKM) (Wu et al., 2018b), interpreted memory writes and reads as inference in a generative model, wherein memory is now treated as a distribution, $p(M)$. Under this framework, memory reads and writes are recast as sampling or updating the memory posterior. The DKM model differs from the KM model by introducing a dynamical addressing rule that could be used throughout training. While providing an intuitive and theoretically sound bound on the data likelihood, the DKM model requires an inner optimization loop which entails solving an ordinary least squares (OLS) problem. Typical OLS solutions require a matrix inversion ($O(n^3)$), preventing the model from scaling to large memory sizes. More recent work has focused on employing a product of smaller Kanerva memories (Marblestone et al., 2020) in an effort to minimize the computational cost of the matrix inversion. In contrast, we propose a simplified view of memory creation by treating memory writes as a deterministic process in a fully feed-forward setting. Crucially, we also modify the read operand such that it uses localized sub-regions of the memory, providing an extra dimension of operation in comparison with the KM and DKM models. While the removal of memory stochasticity might be interpreted as reducing the representation power of the model, we empirically demonstrate through our experiments that our model performs better, trains quicker and is simpler to optimize. The choice of a deterministic memory is further reinforced by research in psychology, where human visual memory has been shown to change deterministically Gold et al. (2005); Spencer & Hund (2002); Hollingworth et al. (2013).

## 3 MODEL

To better understand the K++ model, we examine each of the individual components to understand their role within the complete generative model. We begin by first deriving a conditional variational lower bound (Section 3.1), describing the optimization objective and probabilistic assumptions. We then describe the write operand (Section 3.3), the generative process (Section 3.4) and finally the read and iterative read operands (Section 3.5).

### 3.1 PRELIMINARIES

K++ operates over an exchangeable episode (Aldous, 1985) of samples, $X = \{x_t\}_{t=1}^T \in \mathcal{D}$, drawn from a dataset $\mathcal{D}$, as in the Kanerva Machine. Therefore, the ordering of the samples within the episode does not matter. This enables factoring the conditional, $p(X|M)$, over each of the individual samples: $\prod_{t=1}^T p(x_t|M)$, given the memory, $M \sim p(M), M \in \mathbb{R}^{\hat{C} \times \hat{W} \times \hat{H}}$. Our objective in this work is to maximize the expected conditional log-likelihood as in (Bornschein et al., 2017; Wu et al., 2018a):

$$\mathcal{J} = \mathbb{E}_{p(X),\ p(M|X)} \ln p_\theta(X|M) = \int \int p(X)p(M|X) \sum_{t=1}^T \ln p_\theta(x_t|M) dX dM \qquad (1)$$

As alluded to in Barber & Agakov (2004) and Wu et al. (2018a), this objective can be interpreted as maximizing the mutual information, $I(X;M)$, between the memory, $M$, and the episode, $X$, since $I(X;M) = H(X) + \mathcal{J} = H(X) - H(X;M)$ and given that the entropy of the data, $H(X)$, is constant. In order to actualize Equation 1 we rely on a variational bound which we derive in the following section.

### 3.2 VARIATIONAL LOWER BOUND

To efficiently read from the memory, $M$, we introduce a set of latent variables corresponding to the $K$ addressing read heads, $Y = \{Y_t = \{y_{tk}\}_{k=1}^K\}_{t=1}^T, y_{tk} \in \mathbb{R}^3$ , and a set of latent variables corresponding to the readout from the memory, $Z = \{z_t\}_{t=1}^T, z_t \in \mathbb{R}^L$. Given these latent variables, we can decompose the conditional, $\ln p(X|M)$, using the product rule and introduce variational approximations $q_\phi(Z|X)$ [2] and $q_\phi(Y|X)$ via a multiply-by-one trick:

---

[1] While DNC is slot based, it should be noted that DNC reads rows rather than columns.

[2] We use $q_\phi(Z|X)$ as our variational approximation instead of $q_\phi(Z|X, Y, M)$ in DKM as it presents a more stable objective. We discuss this in more detail in Section 3.5.

$$\ln p(X|M) = \ln \frac{p(X, Z, Y|M)}{p(Z, Y|X, M)} = \ln \frac{p(X|Z, M, Y) \, p(Z|M, Y) \, p(Y|M)}{p(Z|M, Y, X) \, p(Y|M, X)}$$

$$\approx \mathbb{E}_{q_\phi(Z|X), q_\phi(Y|X)} \left( \ln \frac{p(X|Z, M, Y) \, p(Z|M, Y) \, p(Y) \, q_\phi(Z|X) \, q_\phi(Y|X)}{p(Z|M, Y, X) \, p(Y|M, X) \, q_\phi(Z|X) \, q_\phi(Y|X)} \right) \quad (2)$$

$$= \mathcal{L}_T + \underbrace{\mathbb{E}_{q_\phi(Y|X) \, q_\phi(M|X,Y)} \mathcal{D}_{KL}(q_\phi(Z|X)||p(Z|M, Y, X))}_{\geq 0} \quad (3)$$

$$+ \underbrace{\mathcal{D}_{KL}(q_\phi(Y|X)||p(Y|M, X))}_{\geq 0}$$

Equation 2 assumes that $Y$ is independent from $M$: $p(Y|M) = p(Y)$. This decomposition results in Equation 3, which includes two KL-divergences against true (unknown) posteriors, $p(Z|M, Y, X)$ and $p(Y|M, X)$. We can then train the model by maximizing the evidence lower bound (ELBO), $\mathcal{L}_T$, to the true conditional, $\ln p(X|M) > \mathcal{L}_T$:

$$\mathcal{L}_T = \underbrace{\mathbb{E}_{q_\phi(Z|X), \, q_\phi(Y|X)} \ln p_\theta(X|Z, M, Y)}_{\text{Decoder}}$$

$$- \underbrace{\mathbb{E}_{q_\phi(Y|X)} \mathcal{D}_{KL}[q_\phi(Z|X)||p_\theta(Z|M, Y)]}_{\text{Amortized latent variable posterior vs. memory readout prior}} \quad (4)$$

$$- \underbrace{\mathcal{D}_{KL}[q_\phi(Y|X)||p(Y)]}_{\text{Amortized key posterior vs. key prior}}$$

The bound in Equation 4 is tight if $q_\phi(Z|X) = p(Z|M, Y, X)$ and $q_\phi(Y|X) = p(Y|M, X)$, however, it involves inferring the entire memory $M \sim q_\phi(M|X, Y)$. This prevents us from decoupling the size of the memory from inference and scales the computation complexity based on the size of the memory. To alleviate this constraint, we assume a purely deterministic memory, $M \sim \delta[f_{mem}(f_{enc}(X))]$, built by transforming the input episode, $X$, via a deterministic encoder and memory transformation model, $f_{mem} \circ f_{enc}$. We also assume that the regions of memory which are useful in reconstructing a sample, $x_t$, can be summarized by a set of $K$ localized contiguous memory sub-blocks as described in Equation 5 below. The intuition here is that similar samples, $x_t \approx x_r$, might occupy a disjoint part of the representation space and the decoder, $p_\theta(X|\cdot)$, would need to read multiple regions to properly handle sample reconstruction. For example, the digit "3" might share part of the representation space with a "2" and another part with a "5".

$$\hat{M} \sim \delta \left[ \{m_k = f_{ST}(M = f_{mem}(f_{enc}(X)), y_{tk})\}_{k=1}^K \right] \approx q_\phi(M|X, Y) \quad (5)$$

$\hat{M}$ in equation 5 represents a set of $K$ dirac-delta memory sub-regions, determined by the addressing key, $y_{tk}$, and a spatial transformer ($ST$) network Jaderberg et al. (2015), $f_{ST}$ [3]. Our final optimization objective, $\mathcal{L}_T$, is attained by approximating $M \sim q_\phi(M|X, Y)$ from Equation 4 with $\hat{M}$ (Equation 5) and is summarized by the graphical model in 2 below.

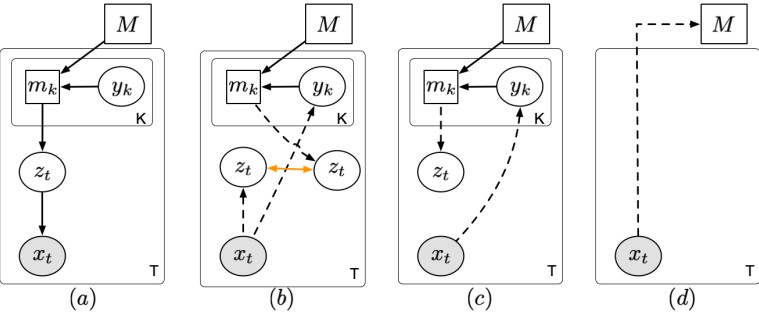

Figure 2: *(a)*: Generative model (§3.4). *(b)*: Read inference model (§3.5). *(c)*: Iterative read inference model (§3.5). *(d)*: Write inference model (§3.3). Dashed lines represent approximate inference, while solid lines represent computing of a conditional distribution. Double sided arrow in (c) represents the KL divergence between $q_\phi(Z|X)$ and $p_\theta(Z|M, Y)$ from Equation 4. Squares represent deterministic nodes. Standard plate notation is used to depict a repetitive operation.

---

[3]We provide a brief review of spatial transformers in Appendix A

### 3.3 WRITE MODEL

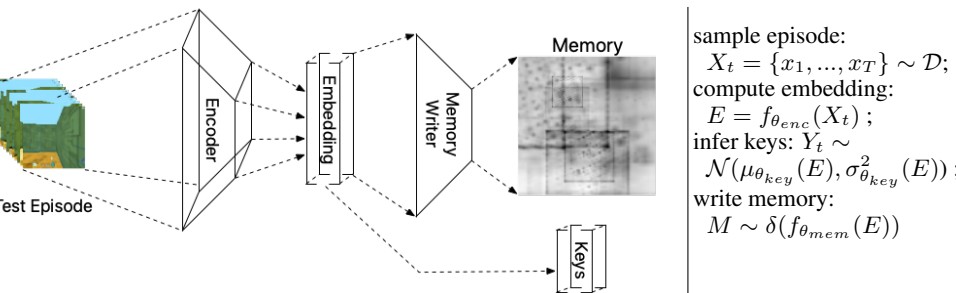

Figure 3: *Left*: Write model. *Right*: Write operation.

Writing to memory in the K++ model (Figure 3) entails encoding the input episode, $X_t = \{x_t\}_{t=1}^T$, through the encoder, $E = f_{\theta_{enc}}(X_t)$, pooling the representation over the episode and encoding the pooled representation with the memory writer, $M = f_{\theta_{mem}}(E)$. In this work, we employ a Temporal Shift Module (TSM) (Lin et al., 2019), applied on a ResNet18 (He et al., 2016). TSM works by shifting feature maps of a two-dimensional vision model in the temporal dimension in order to build richer representations of contextual features. In the case of K++, this allows the encoder to build a better representation of the memory by leveraging intermediary episode specific features. Using a TSM encoder over a standard convolutional stack improves the performance of both K++ and DKM, where the latter observes an improvement of 6.32 nats/image over the reported test conditional variational lower bound of 77.2 nats/image (Wu et al., 2018b) for the binarized Omniglot dataset. As far as we are aware, the application of a TSM encoder to memory models has not been explored and is a contribution of this work.

The memory writer model, $f_{\theta_{mem}}$, in Figure 3, allows K++ to non-linearly transform the pooled embedding to better summarize the episode. In addition to inferring the deterministic memory, $M$, we also project the non-pooled embedding, $E$, through a key model, $f_{\theta_{key}}$:

$$\{Y_t\}_{t=1}^T = \mu_{\theta_{key}}(E) + \sigma_{\theta_{key}}^2(E) \odot \epsilon, \quad y_{tk} \in \mathbb{R}^3, \quad \epsilon \sim \mathcal{N}(0,1). \tag{6}$$

The reparameterized keys will be used to read sample specific memory traces, $\hat{M}$, from the full memory, $M$. The memory traces, $\hat{M}$, are used in training through the learned prior, $p_\theta(Z|\hat{M}, Y) = \mathcal{N}(\mu_z, \sigma_z^2)$, from Equation 4 via the KL divergence, $\mathbb{E}_{q_\phi(Y|X)} \mathcal{D}_{KL}[q_\phi(Z|X)||P_\theta(Z|\hat{M}, Y)]$. This KL divergence constrains the optimization objective to keep the representation of the amortized approximate posterior, $q_\phi(Z|X)$, (probabilistically) close to the memory readout representation of the learned prior, $p_\theta(Z|\hat{M}, Y)$. In the generative setting, this constraint enables memory traces to be routed from the learned prior, $p_\theta(Z|\hat{M}, Y)$, to the decoder, $p_\theta(X|\cdot)$, in a similar manner to standard VAEs. We detail this process in the following section.

### 3.4 SAMPLE GENERATION

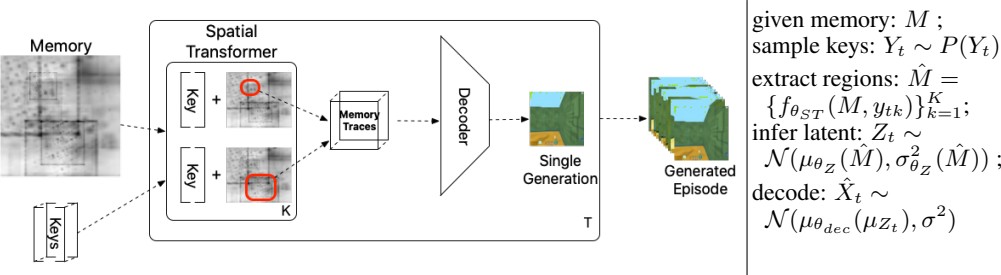

Figure 4: *Left*: Generative model. *Right*: Generative operation.

The Kanerva++ model, like the original KM and DKM models, enables sample generation given an existing memory or set of memories. $K$ samples from the prior key distribution, $\{y_k\}_{k=1}^K \sim p(Y) = \mathcal{N}(0, 1), y_k \in \mathrm{R}^3$, are used to parameterize the spatial transformer, $f_{ST}$, which indexes the deterministic memory, $M$. The result of this differentiable indexing is a set of memory sub-regions, $\hat{M}$, which are used in the decoder, $p_\theta(X|\cdot)$, to generate synthetic samples. Reading samples in this manner forces the encoder to utilize memory sub-regions that are useful for reconstruction, as non-read memory regions receive zero gradients during backpropagation. This insight allows us to use the simple feed-foward write process described in Section 3.3, while still retaining the ability to produce locally contiguous block allocated memory.

## 3.5 READ / ITERATIVE READ MODEL

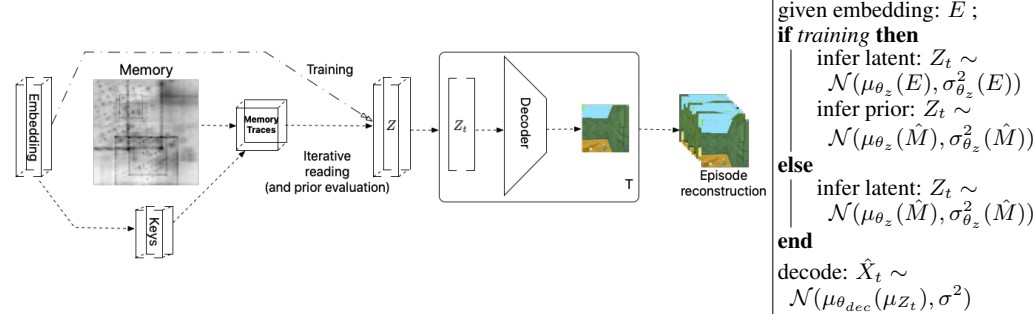

Figure 5: *Left*: Read model: bottom branch from embedding used during iterative reading and prior evaluation. Stable top branch used to infer $q_\phi(Z|X)$ during training. *Right*: Read operation.

K++ involves two forms of reading (Figure 5): iterative reading and a simpler and more stable read model used for training. During training we actualize $q_\phi(Z|X)$ from Equation 4 using an amortized isotropic-gaussian posterior that directly transforms the embedding of the episode, $E$, using a learned neural network (Figure 2-*b*). As mentioned in Section 3.5, the readout, $Z$, of the memory traces, $\hat{M}$, are encouraged to learn a meaningful structured representation through the memory read-out KL divergence, $\mathbb{E}_{q_\phi(Y|X)}\mathcal{D}_{KL}[q_\phi(Z|X)||P_\theta(Z|\hat{M}, Y)]$, which attempts to minimize the (probabilistic) distance between $q_\phi(Z|X)$ and $P_\theta(Z|\hat{M}, Y)$.

Kanerva memory models also possess the ability to gradually improve a sample through interative inference (Figure 2-*c*), whereby noisy samples can be improved by leveraging contextual information stored in memory. This can be interpreted as approximating the posterior, $q(Z|X, M)$, by marginalizing the approximate key distribution:

$$q(Z|X, \hat{M}) = \int q_\phi(Y|X)p_\theta(Z|Y, \hat{M})\delta Y \approx p_\theta(Z|Y = Y^*, \hat{M}), \qquad (7)$$

where $Y^* \sim q_\phi(Y|X = \hat{X})$ in Equation 7 uses a single sample Monte Carlo estimate, evaluated by re-inferring the previous reconstruction, $\hat{X} \sim p_\theta(X|\cdot)$, through the approximate key posterior. Each subsequent memory readout, $Z$, improves upon its previous representation by absorbing additional information from the memory.

## 4 EXPERIMENTS

We contrast K++ against state-of-the-art memory conditional vision models and present empirical results in Table 1. Binarized datasets assume Bernoulli output distributions, while continuous values are modeled by a discretized mixture of logistics Salimans et al. (2017). As is standard in literature Burda et al. (2016); Sadeghi et al. (2019); Ma et al. (2018); Chen et al. (2017), we provide results for binarized MNIST and binarized Omniglot in nats/image and rescale the corresponding results to bits/dim for all other datasets. We describe the model architecture, the optimization procedure and the memory creation protocol in Appendix E and E.1. Finally, extra Celeb-A generations and test image reconstructions for all experiments are provided in Appendix B and Appendix D respectively.

| Method | Binarized MNIST (nats / image) | Binarized Omniglot (nats / image) | Fashion MNIST (bits / dim) | CIFAR10 (bits / dim) | DMLab Mazes (bits / dim) |
|---|---|---|---|---|---|
| VAE Kingma & Welling (2014) | 87.86 | 104.75 | 5.84 | 6.3 | - |
| IWAE Burda et al. (2016) | **85.32** | **103.38** | - | - | - |
| **Improved decoders** | | | | | |
| PixelVAE++ Sadeghi et al. (2019) | 78.00 | - | - | **2.90** | - |
| MAE Ma et al. (2018) | **77.98** | **89.09** | - | 2.95 | - |
| DRAW Gregor et al. (2015) | 87.4 | 96.5 | - | 3.58 | - |
| MatNet Bachman (2016) | 78.5 | 89.5 | - | 3.24 | - |
| **Richer priors** | | | | | |
| Ordered ACN Graves et al. (2018) | $\leq$**73.9** | - | - | $\leq$3.07 | - |
| VLAE Chen et al. (2017) | 78.53 | 102.11 | - | **2.95** | - |
| VampPrior Tomczak & Welling (2018) | 78.45 | **89.76** | - | - | - |
| **Memory conditioned models** | | | | | |
| VMA Bornschein et al. (2017) | - | 103.6 | - | - | - |
| KM Wu et al. (2018a) | - | $\leq$68.3 | - | $\leq$4.37[+] | - |
| DNC Graves et al. (2016) | - | $\leq$100 | - | - | - |
| DKM Wu et al. (2018b) | $\leq$75.3* | $\leq$77.2 | - | $\leq$4.79* | $\leq$**2.75**[†] |
| DKM w/TSM (our impl) | $\leq$51.84 | $\leq$70.88 | $\leq$4.15 | $\leq$4.31 | $\leq$2.92[†] |
| Kanerva++ (ours) | $\leq$**41.58** | $\leq$**66.24** | $\leq$3.40 | $\leq$3.28 | $\leq$2.88[†] |

Table 1: Negative test likelihood and conditional test likelihood values (lower is better). * was graciously provided by original authors. [+] estimated from Wu et al. (2018a) Appendix Figure 12. [†] variadic performance due to online generation of DMLab samples.

K++ presents state-of-the-art results for memory conditioned binarized MNIST and binarized Omniglot, and presents competitive performance for Fashion MNIST, CIFAR10 and DMLab mazes. The performance gap on the continuous valued datasets can be explained by our use of a simple convolutional decoder, rather than the autoregressive decoders used in models such as PixelVAE Sadeghi et al. (2019). We leave the exploration of more powerful decoder models to future work and note that our model can be integrated with autoregressive decoders.

## 4.1 ITERATIVE INFERENCE

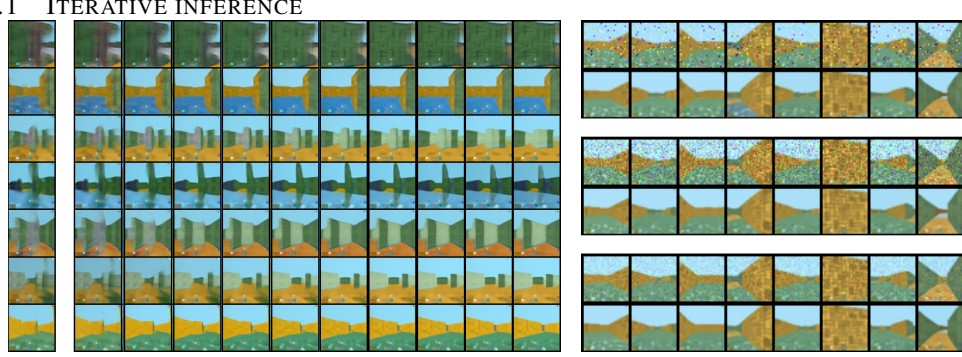

Figure 6: *Left*: First column to left visualizes first random key generation. Following columns created by inferring previous sample through K++. *Right*: denoising of salt & pepper (top), speckle (middle) and Poisson noise (bottom).

One of the benefits of K++ is that it uses the memory to learn a more informed prior by condensing the information from an episode of samples. One might suspect that based on the dimensionality of the memory and the size of the read traces, the memory might only learn prototypical patterns, rather than a full amalgamation of the input episode. This presents a problem for generation, as described in Section 3.4 , and can be observed in the first column of Figure 6-*Left* where the first generation from a random key appears blurry. To overcome this limitation, we rely on the iterative inference of Kanerva memory models (Wu et al., 2018a;b). By holding the memory, $M$, fixed and repeatedly inferring the latents, we are able to clean-up the pattern by leveraging the contextual information contained within the memory (§ 3.5). This is visualized in the proceeding columns of Figure 6-*Left*, where we observe a slow but clear improvement in generation quality. This property of iterative inference is one of the central benefits of using a memory model over a tradition solution like a VAE. We also present results of iterative inference on more classical image noise distributions such as salt-and-pepper, speckle and Poisson noise in Figure 6-*Right*. For each original noisy pattern (top rows) we provide the resultant final reconstruction after ten steps of clean-up. The proposed K++ is robust to input noise and is able to clean-up most of the patterns in a semantically meaningful way.

### 4.1.1 IMAGE GENERATIONS

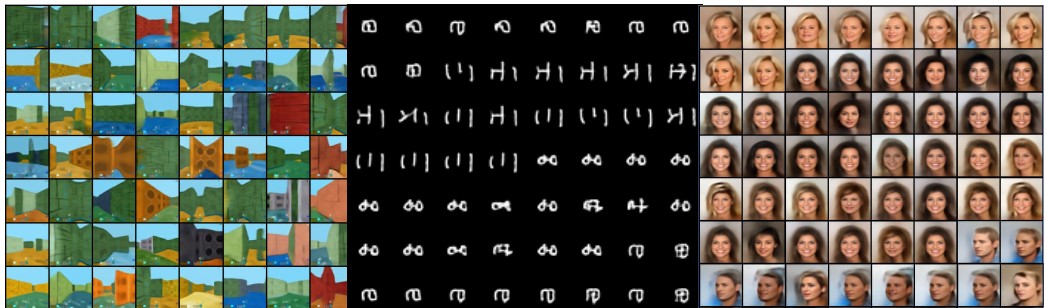

Figure 7: Key perturbed generations. *Left*: DMLab mazes. *Center*: Omniglot. *Right*: Celeb-A 64x64.

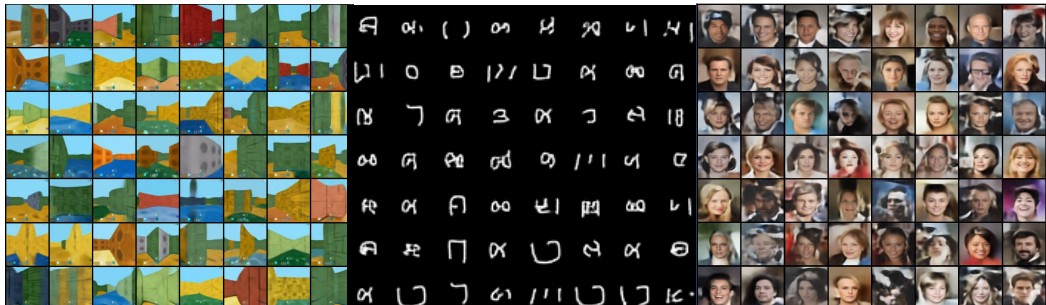

Figure 8: Random key generations. *Left*: DMLab mazes. *Center*: Omniglot. *Right*: Celeb-A 64x64.

Typical VAEs use high dimensional isotropic Gaussian latent variables ($\geq \mathbb{R}^{16}$) Burda et al. (2016); Kingma & Welling (2014). A well known property of high dimensional Gaussian distributions is that most of their mass is concentrated on the surface area of a high dimensional ball. Perturbations to a sample in an area of valid density can easily move it to an invalid density region (Arvanitidis et al., 2018; White, 2016), causing blurry or irregular generations. In the case of K++, since the key distribution, $y_t \sim p(Y), y_t \in \mathbb{R}^3$, is within a low dimensional space, local perturbations, $y_t + \epsilon, \epsilon \sim N(0, 0.1)$, are likely in regions with high probability density. We visualize this form of generation in Figure 7 for DMLab Mazes, Omniglot and Celeb-A 64x64, as well as the more traditional random key generations, $y_t \sim p(Y)$, in Figure 8.

Interestingly, local key perturbations of a trained DMLab Maze K++ model induces resultant generations that provide a natural traversal of the maze as observed by scanning Figure 7-*Left*, row by row, from left to right. In contrast, the random generations of the same task (Figure 8-*Left*) present a more discontinuous set of generations. We see a similar effect for the Omniglot and Celeb-A datasets, but observe that the locality is instead tied to character or facial structure as shown in Figure 7-*Center* and Figure 7-*Right*. Finally, in contrast to VAE generations, K++ is able to generate sharper images of ImageNet32x32 as shown in Appendix C. Future work will investigate this form of locally perturbed generation through an MCMC lens.

### 4.2 ABLATION: IS BLOCK ALLOCATED SPATIAL MEMORY USEFUL?

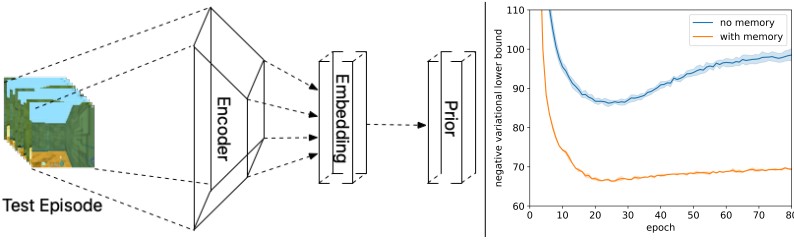

Figure 9: *Left*: Simplified write model directly produces readout prior from Equation 4 by projecting embedding $E$ via a learned network. *Right*: Test negative variational lower bound (mean $\pm$1std).

While Figure 7 demonstrates the advantage of having a low dimensional sampling distribution and Figure 6 demonstrates the benefit of iterative inference, it is unclear whether the performance benefit in Table 1 is achieved from the episodic training, model structure, optimization procedure or memory allocation scheme. To isolate the cause of the performance benefit, we simplify the write architecture from Section 3.3 as shown in Figure 9-*Left*. In this scenario, we produce the learned memory readout, $Z$, via an equivalently sized dense model that projects the embedding, $E$, while keeping all other aspects the same. We train both models five times with the exact same TSM-ResNet18 encoder, decoder, optimizer and learning rate scheduler. As shown in Figure 9-*Right*, the test conditional variational lower bound of the K++ model is **20.6 nats/image** better than the baseline model for the evaluated binarized Omniglot dataset. This confirms that the spatial, block allocated latent memory model proposed in this work is useful when working with image distributions. Future work will explore this dimension for other modalities such as audio and text.

## 4.3 ABLATION: EPISODE LENGTH (T) AND MEMORY READ STEPS (K).

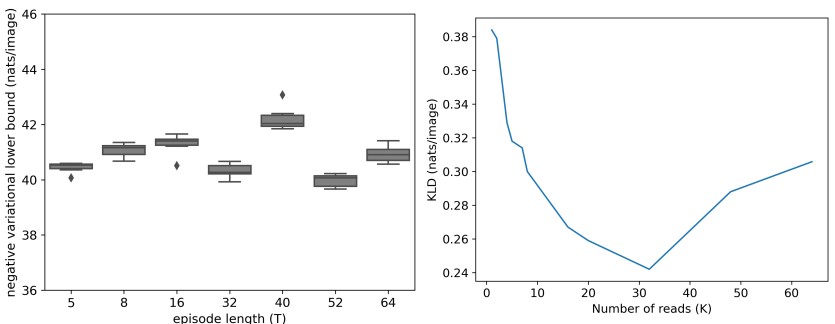

Figure 10: Binarized MNIST. *Left*: Episode length (T) ablation showing negative test conditional variational lower bound (mean±std). *Right*: Memory read steps (K) ablation showing test KL divergence.

To further explore K++, we evaluate the sensitivity of the model to varying episode lengths (T) in Figure 10-*left* and memory read steps (K) in Figure 10-*right* using the binarized MNIST dataset. We train K++ five times (each) for episode lengths ranging from 5 to 64 and observe that the model performs within margin of error for increasing episode lengths, producing negative test conditional variational bounds within a 1-std of $\pm0.625$ nats/image. This suggests that for the specific dimensionality of memory ($\mathbb{R}^{64\times64}$) used in this experiment, K++ was able to successfully capture the semantics of the binarized MNIST distribution. We suspect that for larger datasets this relationship might not necessary hold and that the dimensionality of the memory should scale with the size of the dataset, but leave the prospect of such capacity analysis for future research.

While ablating the number of memory reads (K) in Figure 10-*right*, we observe that the total test KL-divergence varies by 1-std of $\pm0.041$ nats/image for a range of memory reads from 1 to 64. A lower KL divergence implies that the model is able to better fit the approximate posteriors $q_\phi(Z|X)$ and $q_\phi(Y|X)$ to their correspondings priors in Equation 4. It should however be noted that a lower KL-divergence does not necessary imply a better generative model Theis et al. (2016). While qualitatively inspecting the generated samples, we observed that K++ generated more semantically sound generations at lower memory read steps. We suspect that the difficulty of generating realistic samples increases with the number of disjoint reads and found that $K = 2$ produces high quality results. We use this value for all experiments in this work.

## 5 CONCLUSION

In this work, we propose a novel block allocated memory in a generative model framework and demonstrate its state-of-the-art performance on several memory conditional image generation tasks. We also show that stochasticity in low-dimensional spaces produces higher quality samples in comparison to high-dimensional latents typically used in VAEs. Furthermore, perturbations to the low-dimensional key generate samples with high variations. Nonetheless, there are still many unanswered questions: would a hard attention based solution to differentiable indexing prove to be better than a spatial transformer? What is the optimal upper bound of window read regions based on the input distribution? Future work will hopefully be able to address these lingering issues and further improve generative memory models.

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

## A  SPATIAL TRANSFORMER REVIEW

Indexing a matrix, $M[x : x + \Delta x, y : y + \Delta y]$, is typically a non-differentiable operation since it involves hard cropping around an index. Spatial transformers (Jaderberg et al., 2015) provide a solution to this problem by decoupling the problem into two differentiable operands:

1. Learn an affine transformation of coordinates.

2. Use a differntiable bilinear transformation.

The affine transformation of source coordinates, $\begin{bmatrix} i^s \\ j^s \end{bmatrix}$, to target coordinates, $\begin{bmatrix} i^t \\ j^t \end{bmatrix}$ is defined as:

$$\begin{bmatrix} i^t \\ j^t \end{bmatrix} = \begin{bmatrix} s & 0 & x \\ 0 & s & y \end{bmatrix} \begin{bmatrix} i^s \\ j^s \\ 1 \end{bmatrix} = \begin{bmatrix} y_0 & 0 & y_1 \\ 0 & y_0 & y_2 \end{bmatrix} \begin{bmatrix} i^s \\ j^s \\ 1 \end{bmatrix} \tag{8}$$

Here, the affine transform, $\theta = \begin{bmatrix} s & 0 & x \\ 0 & s & y \end{bmatrix}$ has three learnable scalars: $\{s, x, y\}$ which define a scaling and translation in $i$ and $j$ respectively. In the case of K++, these three scalars represent the components of the key sample, $\{y_0, y_1, y_2\} \in \mathbb{R}^3$ as shown in Equation 8. After transforming the co-ordinates (not to be confused with the actual data), spatial transformers learn a differentiable bilinear transform which can be interpreted as learning a differentiable mask that is element-wise multiplied by the original data, $M$:

$$\sum_{n=-1}^{J} \sum_{r=-1}^{J} \left( M_{nr}^c \max(0, 1 - |i_{nr}^t - r|) \max(0, 1 - |j_{nr}^t - n|) \right) \tag{9}$$

Consider the following example where $\theta = \begin{bmatrix} 0.5 & 0 & 0.3 \\ 0 & 0.5 & 0.5 \end{bmatrix}$; this parameterization differntiably extracts the region shown in Figure 11-*Right* from Figure 11-*Left*:

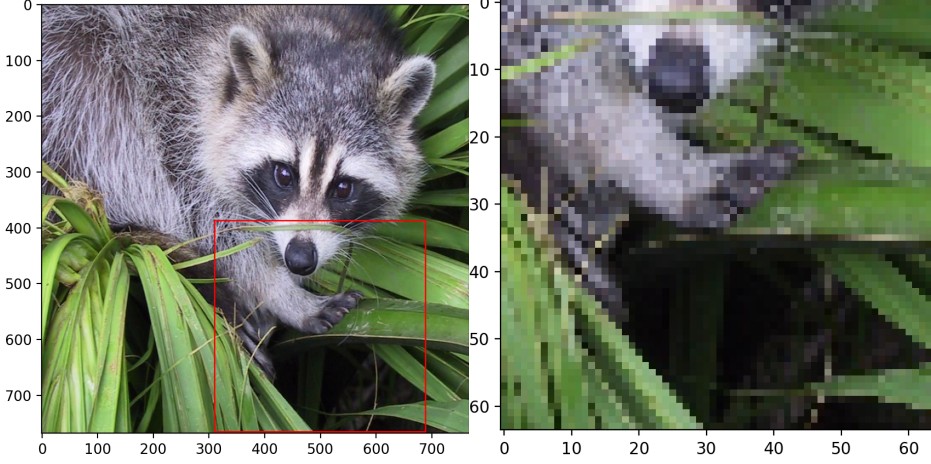

Figure 11: Spatial transformer example. *Left*: original image with region inlaid. *Right*: extracted grid.

The range of values for $\{s, x, y\}$ is bound between $[-1, 1]$, where the center of the image is $[0, 0]$.

## B  CELEB-A GENERATIONS

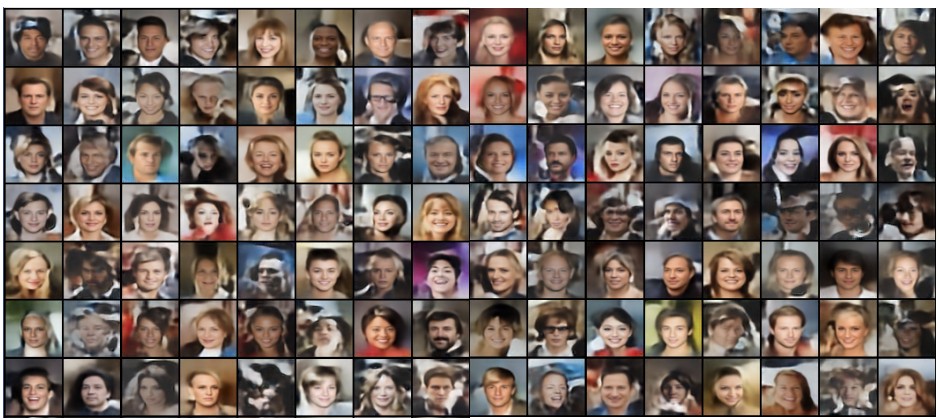

Figure 12: Random key Celeb-A generations.

We present random key generations of Celeb-A 64x64, trained without center cropping in Figure 12.

## C  VAE VS. K++ IMAGENET32X32 GENERATIONS

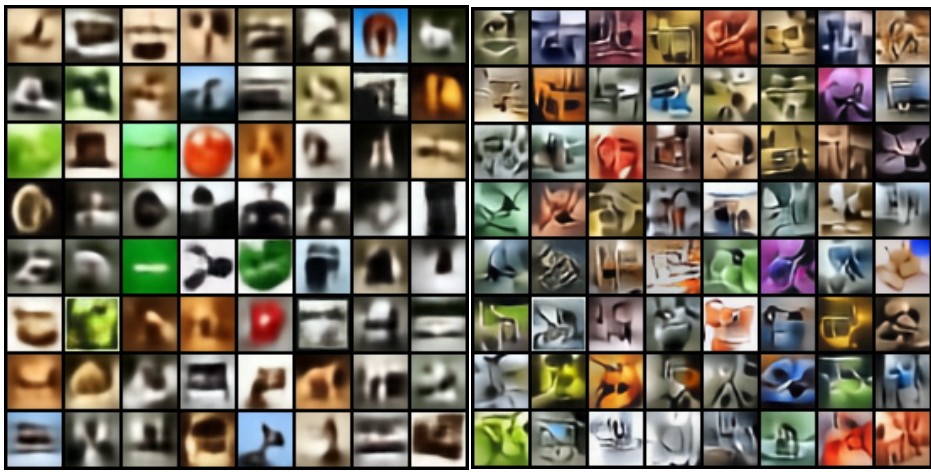

Figure 13: ImageNet32x32 generations. *Left*: VAE; *Right*: K++.

Figure 13 shows the difference in generations of a standard VAE vs. K++. In contrast to the standard VAE generation (Figure 13-*Left*), the K++ generations (Figure 13-*Right*) appear much sharper, avoiding the blurry generations observed with standard VAEs.

## D    TEST IMAGE RECONSTRUCTIONS

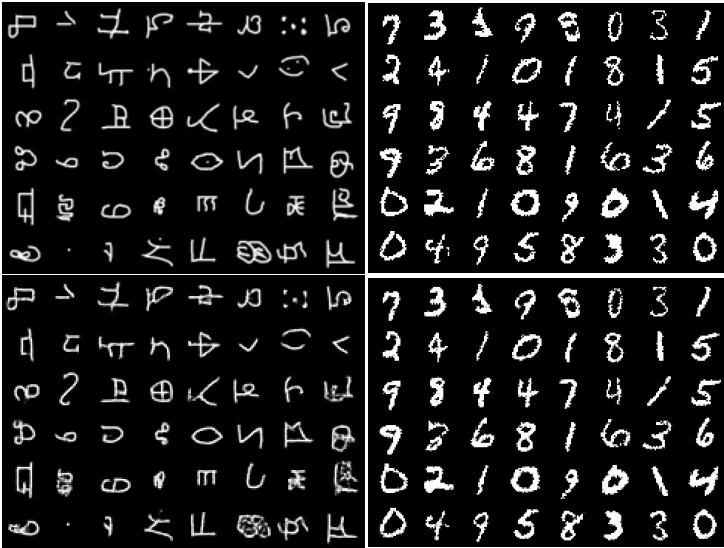

Figure 14: Binarized test reconstructions; top row are true samples. *Left*: Omniglot; *Right*: MNIST.

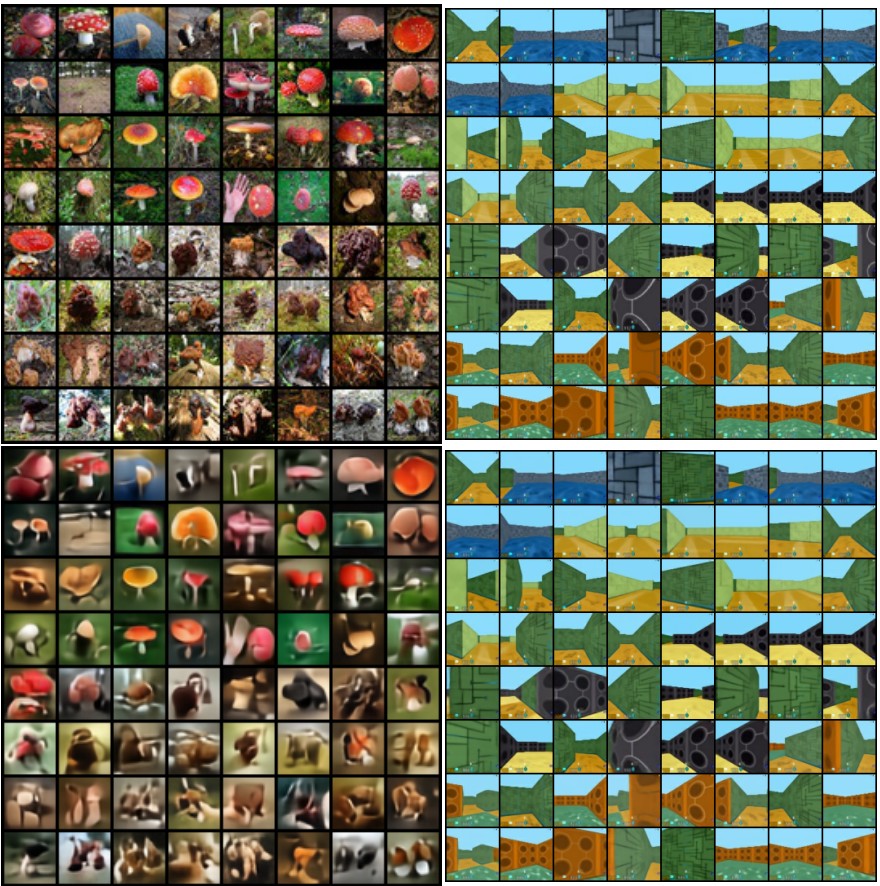

Figure 15: Test set reconstructions; top row are true samples. *Left*: ImageNet64x64. *Right*: DMLab Mazes.

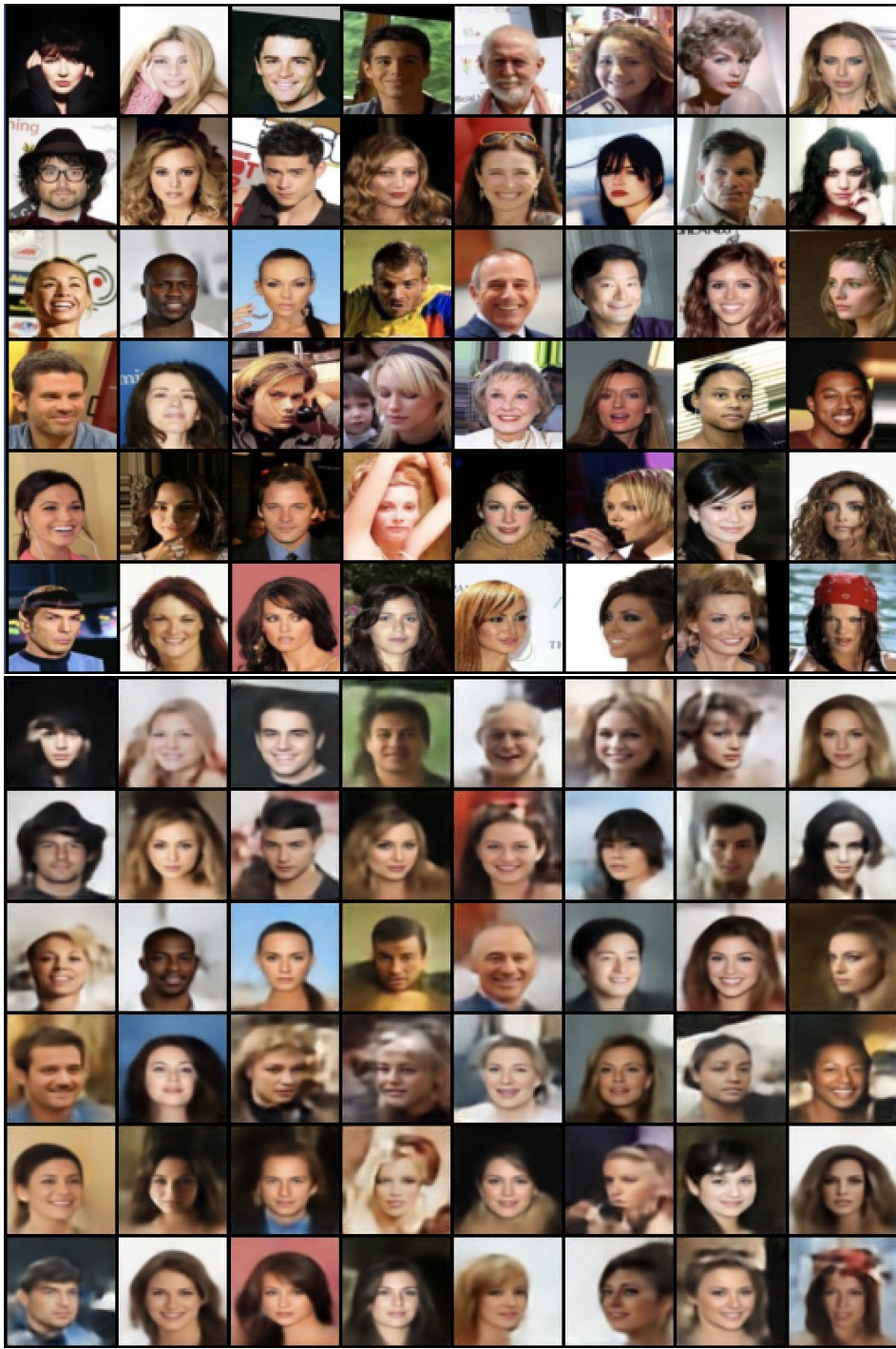

Figure 16: Test set reconstructions; top row are true samples. Celeb-A 64x64.

# E  MODEL ARCHITECTURE & TRAINING

**Encoder**: As mentioned in Section 3.3, we use a TSM-Resnet18 encoder with Batchnorm (Ioffe & Szegedy, 2015) and ReLU activations (Nair & Hinton, 2010) for all tasks. We apply a fractional shift of the feature maps by 0.125 as suggested by the authors.

**Decoder**: Our decoder is a simple conv-transpose network with EvoNormS0 (Liu et al., 2020) inter-spliced between each layer. Evonorms0 is similar in stride to Groupnorm (Wu & He, 2020) combined with the swish activation function (Ramachandran et al., 2018).

**Optimizer & LR scheule**: We use LARS (You et al., 2017) coupled with ADAM (Kingma & Ba, 2014) and a one-cycle (Smith & Topin, 2017) cosine learning rate schedule (Loshchilov & Hutter, 2017). A linear warm-up of 10 epochs (Goyal et al., 2017c) is also used for the schedule. A weight decay of $1e - 3$ is used on every parameter barring biases and the affine terms of batchnorm. Each task is trained for 500 or 1000 epochs depending on the size of the dataset.

**Dense models**: All dense models such as our key network are simple three layer deep linear dense models with a latent dimension of 512 coupled with spectral normalization (Miyato et al., 2018).

**Memory writer**: $f_{\theta_{mem}}$ uses a deep linear conv-transpose decoder on the pooled embedding, $E$ with a base feature map projection size of 256 with a division by 2 per layer. We use a memory size of $\mathbb{R}^{3 \times 64 \times 64}$ for all the experiments in this work.

**Learned Prior**: $p_\theta(Z|\hat{M}, Y)$ uses a convolutional encoder that stacks the $K$ read traces, $\{f_{ST}(M, y_{tk})\}_{k=1}^K$, along the channel dimension and projects it to the dimensionality of $Z$.

In practice, we observed that K++ is about 2x as fast (wall clock) compared to our re-implementation of DKM. We mainly attribute this to not having to solve an inner OLS optimization loop for memory inference.

### E.1 MEMORY CREATION PROTOCOL

The memory creation protocol of K++ is similar in stride to that of the DKM model, given the deterministic relaxations and addressing mechanism described in Sections 3.3 and 3.4. Each memory, $M \sim \delta[f_{mem}(f_{enc}(X))]$, is a function of an episode of samples, $X = \{x_t\}_{t=1}^T \in \mathcal{D}$. To efficiently optimize the conditional lower bound in Equation 4, we parallelize the learning objective using a set of minibatches, as is typical with the optimization of neural networks. As with the DKM model, K++ computes the train and test conditional evidence lower bounds in Table 1, by first inferring the memory, $M \sim \delta[f_{mem}(f_{enc}(X))]$, from the input episode, followed by the read out procedure as described in Section 3.5.

