# OpenReview forum: "Kanerva++: Extending the Kanerva Machine With Differentiable, Locally Block Allocated Latent Memory"
_ICLR.cc/2021/Conference — ICLR 2021 Poster_

### Official Review · AnonReviewer4 · 2020-10-28
**Official Blind Review by reviewer4**

**Rating:** 7
**Confidence:** 3

**Review:**

This paper proposes a new memory mechanism based on the Kanerva Machine inspired by computer heap allocation. It stores the knowledge with several segments, which can be shared between data. The machine stores the knowledge more efficiently through the sharable part-based system and the authors simplified the writing mechanism by designing the memory deterministic.

Strengths:
1. They designed a new Kanerva Machine having a simplified writing mechanism and sharable part-based memory.
2. They evaluated their method on several datasets (e.g., DMLab Mazes or CIFAR10) and showed better performance than previous works.
3. They analyzed local key permutation and ablation study with the version without the memory.

Weaknesses:
1. I think that section 3.3 requires the additional description. As I understand, the memory is updated directly from the input episode, X through f_enc and f_mem, then where is the inferred key used? in the paper, the authors mentioned the key model is described in Section 3.4, but I can't find the description in the section. I think that it is used in the Read inference model, but it is not clearly described.
2. The lack of analysis: this model uses part-based memory, then what happens when the K is decreasing or increasing? I think that the ratio between memory size and the K can be important, when the ratio (K/memory size) is increasing, what happens? What happens with a simplified write mechanism? (I saw you mentioned that you observed K++ is about 2 times faster than DKM, but it is because of the writing mechanism?)

The correctness of their claim and Clarity:

This paper is well-written and correct I think, but if the section 3.3 and 3.5 was written more kindly, then it would be better.

Additional feedback:

Thank you for submitting, I enjoyed reading. Basically, I think the idea of this paper (simplified write mechanism and sharable part-based memory) is good enough and you evaluated it well. However, as I mentioned, for me, some sections are hard to follow and some interesting analyses are not included. And, the discussions about the down-stream tasks like how can we use it for other tasks like model-based RL? also make this paper more concrete.

Minor things are

On page5,
equation 6, R^{T x 3} is right? not T x K x 3?
figure 4, infer latent part, \mu_{\theta_{\hat{M}}}(E) and \sigma^2_{\theta_{\hat{M}}}(E) are right? there is no E in that part.

---

> ### Author Response · Authors · 2020-11-18
> **Clarity improvements as suggested**
>
> We thank the reviewer for their recommendation and hope that our responses below further ratify your decision!
>
>  - Section 3.3 has been expanded as requested to describe the inference of the key and how it is incorporated during training.
>  - Section 3.5 has been updated to describe the read process (training) vs. the iterative read (improving a reconstruction) where the latter affords a probabilistic perspective via the marginalization of the key distribution from the learned prior.
>  - We have added ablations for the episode length (T) and the number of memory reads (K) which hopefully address your request.
>  - K++ is indeed faster due to the write mechanism. DKM has an inner OLS problem that it needs to solve at each step, making it very computational intensive (scales O(n^3) for matrix inversion).
>  - Thanks for the catch on the algorithm for Figure 4! There is indeed no E here. Sampling notation has also been fixed.

---

### Official Review · AnonReviewer3 · 2020-10-28
**An interesting paper on generative memory, yet the writing needs to be improved**

**Rating:** 6
**Confidence:** 4

**Review:**

The paper proposes a generative memory (K++) that takes inspiration from Kanerva Machine and heap memory allocation. An episode of images is encoded into keys and the memory using Temporal Shift Module and transposed convolution. Given the memory and the keys,  a Spatial Transformer reads multiple contiguous memory blocks, which are then used to generate samples using a transposed convolutional decoder.  The model is trained by maximizing the ELBO of the conditional lnP(X|M). The experiments show that K++ can achieve competitive performance on various image datasets.

Pros:
- The idea of using Spatial Transformer to simulate heap memory allocation is interesting
- The performance of the proposed model is very  good for binarized image datasets

Cons:
- The writing is sometimes confusing, hard to follow
- Some details  are missing

Detailed comments and questions:
- The abstract mentions complementary learning system and hierarchical Bayesian memory allocation. However, in the main text, no detail is given to elaborate on these points. How do these concepts embody in the proposed model?
- K++ architecture is related to generative memory models with deterministic memory (especially those that have multiple keys [1,2]). Please make a clear distinction between K++ and those works.
- Eq. 4, How is $q_\phi(Z|X)$ implemented? Fig. 5, How does the memory trace contribute to the read model as mentioned in Sec. 3.4? During training, how are the generative/read models used?
- Eq.6 Writing  $R^{T\times3}$ is inappropriate since $Y_t$ contains $y_{tk}$
- Table 1, reporting two metrics in the same column is misleading
- Table 1, any explanation for the reported values of K++ for CIFAR10? The numbers look strange and use different metric (nats/img), which makes it hard to compare with other baselines.
- Fig. 7, it seems that your model can resist against local perturbation of keys. It is better if you can show that this property is unique to K++. E.g., other models such as Kanerva Machine fails to do so.
- Sec. 4.2 only validates the important role of the memory. To verify the benefit of block memory allocation, you should compare it with other memory indexing schemes. The simplest baseline is to use the whole memory: $\hat{M} = M$ or to read from M using  multi-head slot-based attention (see [1,2]).
- What are the values of T and K used in the experiments?
- Possible writing/syntax error:
in abstract "heirarchar" -->"hierarchical ";
Fig. 4's equations, $Y_t\sim P(Y_t)$ --> $y_t\sim P(Y_t)$;
Fig. 4's equations, given only the memory and keys, where does E come from? It is better to draw Z in the generative model picture;
page 4, Jaderberg et al. (2015) --> (Jaderberg et al., 2015);
page 7,  Burda et al. (2016) --> (Burda et al., 2016);

I like the idea of the paper. However, I feel that the writing is a bit rushed and need revision. Hence, given the current manuscript, I can only give a marginal score. If the authors can address my concerns, improve the writing, explain unclear parts, I will raise my score.

[1]  Mevlana Gemici, Chia-Chun Hung, Adam Santoro, Greg Wayne, Shakir Mohamed, Danilo J Rezende, David Amos, and Timothy Lillicrap. Generative temporal models with memory. arXiv preprint arXiv:1702.04649, 2017.

[2] Hung Le, Truyen Tran, Thin Nguyen, and Svetha Venkatesh. "Variational memory encoder-decoder." In Advances in Neural Information Processing Systems, pp. 1508-1518. 2018.

---

> ### Author Response · Authors · 2020-11-18
> **Improved writing + clarity**
>
> We thank the reviewer for their detailed review and hope to address all your points below (and in the new revision):
>
>   - CLS & Bayesian memory allocation: we have elaborated this in the introduction, specifically "Memory is a central tenant in the model of human intelligence and is crucial to long-term reasoning and planning. Of particular interest is the theory of complementary learning systems McClelland et al. (1995) which proposes that the brain employs two complementary systems to support the acquisition of complex behaviours: a hippocampal fast learning system that records events as episodic memory, and a neocortical slow learning system that learns statistics across events as semantic memory. While the functional dichotomy of the complementary systems are well-established McClelland et al. (1995); Kumaran et al. (2016), it remains unclear whether they are bounded by different computational principles. In this work we introduce a model that bridges this gap by showing that the same statistical learning principle can be applied to the fast learning system through the construction of a hierarchical Bayesian memory."
>   - We have added the requested references and have briefly discussed them. While definitely relevant, both models still fall under the slot based writing mechanism ($z = r^T M$), while K++, KM and DKM describe a hierarchal Bayesian memory model.
>   - $q(Z|X)$: all distributions in our model are simple isotropic gaussians; this particular distribution's parameters ($\mu$, $\sigma^2$) are the outputs of a neural network on the embedding, $E$, of the input data (the output of the TSM model). Also note that Section 3.5 was restructured to explain reading (training) vs. iterative reading (using memory to improve reconstructions) with an interpretation of the latter as marginalizing the key distribution from the learned prior. The memory traces are used in the KL divergence from Equation 4. They attempt to keep the output of the memory readout, $Z$, close to the representation extracted by the data ( $q(Z|X)$ ). During training we do not use the generative or iterative reading steps from the graphical model. They are only used to generate images and iteratively cleanup images (respectively).
>   - Table 1: current literature seems to favor reporting nats/image for binarized image datasets, while bits/dim is used for continuous valued datasets (see [1, 2, 3] for example) where the latter is a simple scaling of the former, i.e.: bits/dim = likelihood / dimensions_of_input / ln(2). We have also fixed the CIFAR10 reporting and have used a more standard method of reporting likelihoods that is consistent with current literature.
>   - Resistance to perturbation: while the Kanerva machine would also be resistant to local perturbations, it would require that the read-key be housed in a low dimensional space. In turn this would require the memory of the Kanerva machine to also be in the [low_dimension x low_dimension] space, thus reducing its performance. In contrast, our model decouples the size of the memory from the size of the read key, allowing perturbations  to be lie within a feasible density region. This is not feasible in a high dimensional space due to the curse of dimensionality.
>   - Memory indexing schemes: the reviewer makes a good point, but we would argue that KM, DKM and DNC are exactly that baseline (they use the entire memory) and we show that K++ improves upon those baselines.
>   - We set T=32 for the binarized datasets and T=18 for the continuous valued datasets. K was set to 2 for all experiments. We have also added ablations to expand upon this further.
>   - E is the output of the TSM encoder and is deterministic given the input episode. In essence it takes a set of images, [B, T, C, H, W], where B = batch size, [C, H, W] are the image dimensions, and projects them to [B, T, F], where F is a feature dimension.
>   - Where to draw Z from: Z is drawn from $p_{\theta}(Z|M, Y)$ when generating synthetic data. This is similar in stride to how VAEs generate synthetic samples after training. During training $p_{\theta}(Z|M, Y)$ is kept (probabilistically) close to $q_{\phi}(Z|X)$. We have drastically improved the clarity of Section 3.5 and hope the reviewer takes a glance!
>
> References:
>
> [1] Ishaan Gulrajani, Kundan Kumar, Faruk Ahmed, Adrien Ali Täıga, Francesco Visin, David Va ́zquez, and Aaron C. Courville. Pixelvae: A latent variable model for natural images. ICLR 2017.
> [2] Xi Chen, Diederik P Kingma, Tim Salimans, Yan Duan, Prafulla Dhariwal, John Schulman, Ilya Sutskever, and Pieter Abbeel. Variational lossy autoencoder. ICLR, 2017.
> [3] Philip Bachman. An architecture for deep, hierarchical generative models. NeuRIPS 2016.

---

### Official Review · AnonReviewer2 · 2020-10-28
**Clarity of the description and experiments can be improved.**

**Rating:** 6
**Confidence:** 4

**Review:**

This paper proposes a generative memory modeling method.  Specifically, this paper proposed a novel memory allocation scheme,  replacing the stochastic memory writing process in prior works KM[1]  and  DKM[2] with a set of deterministic operations. This deterministic process is implemented as the spatial transformation[3] on the pooled embedding space of input data.  The authors showed the proposed model achieved state-of-the-art results on the binarized MNIST data set and the binarized Omniglot data set.  The authors also showed that introducing the work in TSM[4] helps learn a richer context representation compared with 2D convolution layers.

Summary of positive:
The method integrated with TSM and showed improvement on DKM.
The quantitative experiment result shows a great improvement on previous work.


Summary of negative:
The method section is not very clear to me, e.g. how is the key used in the writing process. The reading inference process seems not involved with the memory (section3.5 and figure 5), which is not consistent with the graphic model in figure 2.
Table 1 shows the result of one of the key experiments, however, it is not well-formatted, and some measurements are not well aligned or not completed.  For example, the result on CIFAR10 is reported as nats/img while the baseline models report bits/dim.
The baseline model in the ablation study is too weak. Is it possible to compare with KM with multiple addressing keys? The description of the baseline model is not clear, from the text, it says the learned memory prior (p(z|y,M)?) was replaced with a dense layer. While compared with the figure in section 3.3, it seems the memory writing (f_{mem}) was replaced.


Others:
How is the value K set, and how is it affecting the performance?
Is there any analysis on how the possibly overlapped sub-regions provide a better-compressed representation than the sparse distributed memory?

---

> ### Author Response · Authors · 2020-11-18
> **Clarity improvements**
>
> We appreciate the reviewer's valuable feedback. To address your concerns:
>
>   - Key+writing: the key is not used during writing and is only used during generation / iterative reading (updated in Sec 3.5 + Sec 3.3). The keys parameterize contiguous blocks which are used in the prior (and incorporated in learning via the KL divergence from Eqn 4). Since non-read regions receive 0 gradients, we can directly achieve block allocated memory without writing to the memory in a block fashion.
>   - Graphical model: we have cleaned this up (Fig 2) and have updated Section 3.5 to clearly disambiguate reading (stable for training) from iterative reading (leverages memory for iterative cleanup). We also add a probabilistic interpretation in the same section that explains iterative memory reading as marginalizing of the key distribution from the learned prior.
>   - CIFAR10: we have cleaned this up; we computed the reconstruction using a discretized mixture of logistics and have properly rescaled the results to bits/dim. We now report results in a simplified manner that is used throughout current literature.
>   - Ablation: we have improved  the language for the existing ablation and added two new studies focusing on T and K.
>   - KM w/ multiple keys: since KM uses the entire memory it does not theoretically need multiple keys.

---

### Official Review · AnonReviewer1 · 2020-10-29
**An extension of Mar**

**Rating:** 6
**Confidence:** 4

**Review:**

Summary:
The authors develop a hierarchical Bayesian memory allocation scheme to bridge the gap between episodic and semantic memory via a hierarchical latent variable model. They take inspiration from traditional heap allocation and extend the idea of locally contiguous memory to the Kanerva Machine, enabling a novel differentiable block allocated latent memory. In contrast to the Kanerva Machine, the authors simplify the process of memory writing by treating it as a fully feed forward deterministic process, relying on the stochasticity of the read key distribution to distribute information within the memory.

Pros:
- The authors combine the idea of differentiable indexing in Spatial Transformer (Jaderberg et al., 2015) into the memory of Kanerva Machine (Wu et al, 2018a;b) and prove by experiments that this allocation scheme on the memory helps improve the test negative likelihood. Also, its speed is about 2x faster than the Dynamic Kanerva Machine.
- The authors show the efficiency of Temporal Shift Module (TSM) (Lin et al., 2019) in the encoder of memory models. Replacing a standard convolutional stack by TSM improves the ELBO in Dynamic Kanerva Machine by 6.32 nats/image for the Omniglot dataset.
- The experiments are well reported for different tasks, such as reconstruction and generation, on various datasets.

Cons:
- The whole article is just like a mechanical mixture of old ideas. To be specific, the K++ model is Kanerva Machine + Spatial Transformer + a powerful encoder (namely, Temporal Shift Module). The authors do not introduce any significant improvement or novel insight for old models and techniques.
- Is there any theory support for the idea that we should use an allocated deterministic memory instead of a full variational memory? The authors mention the theory of complementary learning system in the Abstract and heap allocation at the beginning of Section 1, but there is no further analysis for these two theoretical intuitions.

Comments and questions:
- The basic idea is clear and reasonable, but the authors should provide deeper analysis as well as deeper insights for their new model (and for old models that they use, if possible).
- The authors use q_phi(Z|X) in the ELBO (eq. 4) instead of q_phi(Z|X, Y, M) as in the Kanerva Machine. How is the memory used in the read model?
- Where do the results in Table 1 come from? For example, in Kanerva Machine and Dynamic Kanerva Machine paper (Wu et al, 2018a;b), the authors did not report the negative likelihood for CIFAR10 dataset.
- Is there any explanation for the significantly low negative likelihood (-2344.5 bits/dim) of K++ for CIFAR10 dataset?
- The authors should include a brief introduction section about Kanerva Machine and Dynamic Kanerva Machine. Moreover, the function δ in eq. 5 is not defined beforehand.


REFERENCES
James L McClelland, Bruce L McNaughton, and Randall C O’Reilly. Why there are complementary learning systems in the hippocampus and neocortex: insights from the successes and failures of connectionist models of learning and memory. Psychological review, 102(3):419, 1995.
Max Jaderberg, Karen Simonyan, Andrew Zisserman, et al. Spatial transformer networks. In Advances in neural information processing systems, pp. 2017–2025, 2015.
Yan Wu, Greg Wayne, Alex Graves, and Timothy Lillicrap. The Kanerva machine: A generative distributed memory. ICLR, 2018a.
Yan Wu, Gregory Wayne, Karol Gregor, and Timothy Lillicrap. Learning attractor dynamics for generative memory. In Advances in Neural Information Processing Systems, pp. 9379–9388, 2018b.
Ji Lin, Chuang Gan, and Song Han. TSM: Temporal shift module for efficient video understanding. In Proceedings of the IEEE International Conference on Computer Vision, pp. 7083–7093, 2019.

---

> ### Author Response · Authors · 2020-11-18
> **Novel Bayesian spatial memory**
>
> We thank the reviewer for their diligent review and time spent thoroughly reading our work. We hope our responses below convince the reviewer of the merit of this line of research:
>
>   - Mixture of ideas: while the reviewer is correct in that K++ is Kanerva + Spatial Transformer + TSM, we would like to add that it is clearly distinct from each of them and proposes a novel mechanism for differentiable allocation of memory in a manner similar to heap allocators (an idea that has not been explored in prior literature). Spatial transformers for example are not stochastic and operate on input images; in contrast K++ keys are stochastic and handle addressing in latent space. TSM has mainly been used in video models, but we empirically demonstrate that it is very useful in learning to build a common memory pool using contextual information from neighboring features. Finally, while our model is highly related to KM, we are faster (~2x), simpler to implement and optimize and perform better on numerous tasks. At the end of the day, we all stand on the shoulders of giants :)
>   - Complementary learning systems, deterministic memory and insights: we have added more information about CLS in the introduction as well as the following in the related works section to suggest why deterministic memory might be a good idea: "The choice of a deterministic memory is further reinforced by research in psychology, where human visual memory has been shown to change deterministically Gold et al. (2005); Spencer & Hund (2002); Hollingworth et al. (2013)." We have also added numerous insights across the paper such as the reason for multiple reads (K): "The intuition here is that similar samples, $x_t ≈ x_r$, might occupy a disjoint part of the representation space and the decoder, $p_{\theta}(X|·)$, would need to read multiple regions to properly handle sample reconstruction. For example, the digit “3” might share part of the representation space with a “2” and another part with a “5”."
>   - $q_{\phi}(Z|X)$: we have elaborated Section 3.5 to describe the difference between reading (stable & used for training) and iterative reading (leverages memory to improve reconstructions and can be interpreted as marginalizing the key distribution from learned prior, [Sec 3.5 for maths] ). Note that the choice of the variational approximation, $q_{\phi}(Z|X)$, is a choice (it is introduced via a multiply by 1, so any distribution can be chosen), and we choose this specific distribution as it enables stable training.
>   - The results for KM & DKM (now properly explained in paper) come from Appendix Figure 12 of KM and graciously provided by the original authors for DKM.
>   - CIFAR10: we have rectified this by using discretized mixture of logistics to fairly contrast all other models and have standardized the notation in a manner consistent with current literature.
>   - Info about KM & DKM expanded in related works section.
>   - $\delta$ properly described as dirac-delta.

---

### Author Response · Authors · 2020-11-18
**TLDR Revision Changes**

Key changes:

  1. Clarify read (training) vs. iterative read (uses memory) in Section 3.5 and improve Section 3.3 (explaining how keys are computed and used).
  2. Standardize likelihood table in a manner consistent with current literature.
  3. FashionMnist, CIFAR10 & DMLab mazes properly reported using discretized mixture of logistics to accurately constrast all models.
  4. Improve introduction, motivate with more complementary learning systems research as well as suggest why heap allocation and deterministic memory is a good idea.
  5. Add episode length ablation (T) and number of memory reads (K) ablation.
  6. Cleaner graphical model (Fig 2) which shows clear difference between read / iterative read, generative and write processes.

Rev 1.01:
  - Minor fixes including typos and minimal notation improvements.

Rev 1.1:
  - Add memory readout z ~ p(Z|M, Y) and z ~ q(Z|X) to graphical model in Figure 2.
  - Episode ablation now includes variance bars.
  - Explain that Equation 5 is approximation q(M | X, Y) which is used in L_T wherever there is an M.

---

### Decision · Program_Chairs · 2021-01-07
**Final Decision**

**Decision:**

Accept (Poster)

**Comment:**

The paper proposes a variant of Kanerva Machine Wu et al. (2018) by introducing a spatial transformer to index the memory storage and Temporal Shift Module Lin et al., (2019). The KM++ model learns to encode an exchangeable sequence locally via the spatial transformer. The proposed method is evaluated on conditional image generation tasks. The empirical results demonstrated the nearby keys in the memory encoded related and similar images. Several issues of clarity and the correctness of the main theoretical result were addressed during the rebuttal period in a way that satisfied the reviewers. The basic ideas in the paper are interesting to both the machine learning and the wider cognitive science communities. However, additional experiments should be included in Table 1 to complete the "DKM w/TSM (our impl)" row on Fashion MNIST, CIFAR-10, and DMLab in the final revision for completeness.